# Salt-Specific Suppression of the Cold Denaturation of Thermophilic Multidomain Initiation Factor 2

**DOI:** 10.3390/ijms24076787

**Published:** 2023-04-05

**Authors:** Veronika Džupponová, Nataša Tomášková, Andrea Antošová, Erik Sedlák, Gabriel Žoldák

**Affiliations:** 1Department of Biophysics, Faculty of Science, P. J. Šafárik University, Jesenna 5, 04001 Košice, Slovakia; 2Department of Biochemistry, Faculty of Science, P. J. Šafárik University in Košice, Moyzesova 11, 04001 Košice, Slovakia; 3Department of Biophysics, Institute of Experimental Physics, Slovak Academy of Sciences, Watsonova 47, 04001 Košice, Slovakia; 4Center for Interdisciplinary Biosciences, Technology and Innovation Park P.J. Šafárik University, Trieda SNP 1, 04011 Košice, Slovakia; 5Center for Interdisciplinary Biosciences, Cassovia New Industry Cluster, Trieda SNP 1, 04011 Košice, Slovakia

**Keywords:** initiation factor 2, thermodynamic stability, cold denaturation, domain cooperativity, electrostatic interactions

## Abstract

Thermophilic proteins and enzymes are attractive for use in industrial applications due to their resistance against heat and denaturants. Here, we report on a thermophilic protein that is stable at high temperatures (*T_trs, hot_* 67 °C) but undergoes significant unfolding at room temperature due to cold denaturation. Little is known about the cold denaturation of thermophilic proteins, although it can significantly limit their applications. We investigated the cold denaturation of thermophilic multidomain protein translation initiation factor 2 (IF2) from *Thermus thermophilus*. IF2 is a GTPase that binds to ribosomal subunits and initiator fMet-tRNA^fMet^ during the initiation of protein biosynthesis. In the presence of 9 M urea, measurements in the far-UV region by circular dichroism were used to capture details about the secondary structure of full-length IF2 protein and its domains during cold and hot denaturation. Cold denaturation can be suppressed by salt, depending on the type, due to the decreased heat capacity. Thermodynamic analysis and mathematical modeling of the denaturation process showed that salts reduce the cooperativity of denaturation of the IF2 domains, which might be associated with the high frustration between domains. This characteristic of high interdomain frustration may be the key to satisfying numerous diverse contacts with ribosomal subunits, translation factors, and tRNA.

## 1. Introduction

Thermophilic bacterial proteins and enzymes are attractive to the academic sector and for industrial applications [1,2]. Thermophilic proteins and enzymes display high thermal stability and significant activity at elevated temperatures. The high stability and activity of thermophilic proteins at elevated temperatures enable conducting experiments and observing reactions that do not occur at room temperature (e.g., polymerase chain reaction). The molecular mechanism of thermostability has been intensively investigated [3,4,5,6,7]. In quantitative terms, it is expressed by a protein stability curve that describes how the free energy of the denaturation, Δ*G*, is dependent on temperature and how the denaturation enthalpy and entropy change over an extensive temperature range. In the case of proteins, denaturation enthalpy is strongly temperature-dependent due to the high heat capacity change in the process. A significant positive change in heat capacity has an interesting consequence for the depiction of protein stability as a curve, which adopts a parabolic shape. Hence, two different denaturation processes can be observed: one on the hot side and one on the cold side—so-called cold denaturation. The molecular origins of cold denaturation lie in the temperature-dependent hydration of hydrophobic groups, which becomes more favorable at low temperatures [8]. Cold denaturation is not often observed, mainly due to two interrelated reasons. First, cold denaturation usually occurs at temperatures well below the experimental window of spectroscopic/calorimetric techniques. To observe cold denaturation at ambient temperatures, the protein stability curve, Δ*G* vs. *T*, should be shifted horizontally to higher temperatures, indicating a higher hot denaturation transition temperature. Thus, thermophilic proteins can be good candidates for observing cold denaturation.

However, the presence of residual structure in the unfolded state may decrease heat capacity [9,10] and, hence, reduce the curvature of the protein stability curve, which then shifts the threshold for cold denaturation to temperatures that are not readily achievable. Second, the changes in heat capacity are often not sufficiently high to allow observation of cold denaturation. As heat capacity changes are correlated with the number of residues [11] or, more precisely, with the size of the cooperative unit, cold denaturation should be reversible and readily detectable for larger proteins. On the other hand, larger proteins are often arranged in domains (beads on a string or discontinuous domains) that may independently fold/unfold, diminishing the size of the cooperative unit and, hence, the magnitude of the heat capacity change. Taken together, there are only limited cases under which cold denaturation can be observed. 

Translation initiation factor 2 (IF2, Figure 1) is among the largest nonribosomal GTPase involved in initiating protein biosynthesis in the cell. It belongs to the family of structurally related enzymes catalyzing GTP hydrolysis during the early processes of mRNA translation. IF2 plays a central role in translation initiation via the specific selection of fMet-tRNA^fMet^. IF2 specific selection is based on recognition of the N-formyl-methionine and aminoacyl acceptor stem of fMet-tRNA^fMet^ [12,13] and significantly faster association of the 50S subunit and correctly assembled 30S complex, which contains all functional components for successful translation initiation. IF2 also participates in other cellular processes, such as transcription [14], the protein export reaction [15], and the prevention of protein aggregation [16], and it functions as a cellular metabolic sensor that toggles between an active, GTP-bound form under normal conditions and an inactive form with bound ppGpp—a nucleotide messenger produced in bacteria during their starvation [17]. 

For successful translation initiation and recovery, IF2 interacts with many biological partners and undergoes functional cycling, including GTP hydrolysis, GDP/GTP exchange, initiator tRNA binding, and binding to the 30S ribosome subunit. While these activities are facilitated by different domains of IF2, they all act in an orchestrated, sequential manner that is indicative of allosteric-like domain–domain communication at individual interaction stages. Our understanding of the function–structure relationship has been largely improved by structural data. First, the full-length three-dimensional structure of IF2 was resolved for IF2/eIF5B from *M. thermoautotrophicum* [18]. This remarkable structure displays a novel and unusual chalice-like protein fold with two domains separated by a 4 nm long α-helix. This fold possesses some common properties with the domain architectures of EF-Tu and EF-G, suggesting that all three proteins may interact similarly.

Interestingly, the structure of IF2 does not explain how the signal of GTP hydrolysis in the G-domain is propagated to the distal C2 domain. In 2005, a cryo-EM model of ribosome-bound IF2 from *E. coli* was published [19], in which IF2 undergoes conformational rearrangement upon binding to the ribosome. In particular, the authors observed that the portion attributable to the GTP-binding domain (G domain) of IF2 was significantly larger than the G domain of the reoriented crystal structure [19]. Hence, IF2 undergoes large conformational changes during translation initiation. In 2016, 11 cryo-EM structures of the *Thermus thermophilus* translation preinitiation complex were published [12]. The complex contains 30S–IF1–IF2–IF3–mRNA–tRNA. Again, the authors observed that the conformation of ribosome-bound IF2 differs from the crystal structure of *Thermus* IF2 [20]. Similar conformational rearrangements of IF2 were also reported in recent cryo-EM structures of the 70S initiation complex from *Pseudomonas aeruginosa*, with a bound compact form of IF2–GDP and initiator tRNA [21]. NMR and circular dichroism of *E. coli* IF2 revealed a flexible linker connecting the N-terminal and central segments of the protein with the C-terminal domain [22]. Thus, multidomain IF2 may be considered flexible, with the capacity to undergo dynamic conformational transitions. Interestingly, the structure of release factor 2, another translation factor, also exhibits differences between its free form in solution compared with the conformation of its ribosome-bound form [23,24,25,26]. 

The X-ray crystal structures of full-length *Thermus thermophilus* apo IF2 and its complex with GDP have been obtained at 3.1 Å resolution [20,27]. The structure of IF2 bears a resemblance to beads on a string, rather than exhibiting a chalice shape similar to eIF5B. IF2 adopts a stable and ordered conformation when interacting with the ribosome, which is facilitated by the predetermined inherent flexibility and expandability of the protein (Figure 1). Possible elements of flexibility are long helices. The overall dimensions of IF2 depend on the nucleotide status of IF2. The apoform of *T. thermophilus* IF2 is more extended, while the GDP-bound IF2 appears more compact. However, IF2 is significantly larger when in a functionally active state, bound to the 30S ribosomal subunit. Unlike all other translational GTPases, IF2 does not have an effecter domain that stably contacts with the switch II region of the GTPase domain. In summary, cryo-EM reconstructions, NMR experiments, and the 3D structure in solution have shown that IF2 transitions from being flexible in solution to having an extended conformation with disrupted interdomain contacts when interacting with ribosomal complexes.

In our previous work, we analyzed the thermodynamic properties of the reversible denaturation of IF2 along with its isolated G and C fragments [28]. For the purpose of clarity, we used the term “fragments” for proteins obtained upon proteolytical cleavage of IF2. Previously, these fragments were called domains, which is inaccurate based on the structural information that is now available. Circular dichroism in the far-UV region has shown that IF2 and its fragments are well folded. Sedimentation velocity experiments reveal that full-length IF2, the G fragment, and most of the C fragment population are fully monomeric.

Thermal denaturation of multidomain IF2 wt occurs at a high melting temperature of 94.5 °C and displays near-complete reversibility. IF2 can be stabilized by GDP and its analog GTP, as indicated by the higher transition temperatures. The high reversibility of thermal denaturation of such a large (63 kDa) protein is remarkable. This might be a possible consequence of the highly modular multidomain structure of IF2. Thermal denaturation is also found to be highly reversible for the G fragment but not the C fragment. The C fragment of IF2 vividly and irreversibly aggregates upon heating at temperatures above the melting temperature. This adverse aggregation can be suppressed using a denaturing agent, which also restores the high reversibility of thermal denaturation. A similar effect of GndHCl on the reversibility of thermal transition was observed for the *B. stearothermophilus* C fragment [29]. The isothermal unfolding experiment revealed an unusually small m-value (m = d ΔG/d[denaturant]), possibly due to unfolding intermediate(s) and because of independent, spectroscopically overlapping unfolding transitions of the G and C fragments. The observed effects follow the Hofmeister series of salts or others. Here, we evaluate the effect of salts on the cold denaturation of thermophilic multidomain initiation factor 2 in the hope of determining the dominant contributions to cold denaturation and to the stability of IF2 and its domains. A possible major contributor could be the screening of electrostatic interactions of IF2 domains by salts (lower dependence on salt type) or nonelectrostatic components and dispersion forces (strongly dependent on salt type). By comparing the effect of chloride salts of the cations Li^+^, Na^+^, NH^4+^, and Cs^+^ and sodium perchlorate on thermodynamic values, we found that the suppression of cold denaturation is highly salt-specific.

## 2. Results

### 2.1. Cold and Hot Denaturation Processes of IF2 Are Sensitive to Salts

Circular dichroism in the far-UV region is a highly sensitive technique for studying the conformational stability of proteins. This is due to the strong absorption of the peptide group and the high sensitivity of electron transitions n→π* (at 222 nm) and π→π * (at 208 nm) to the protein secondary structure [30]. Circular dichroism at 222 nm was used to monitor the thermal unfolding of IF2 in the presence of 0.1 M NaCl and a phosphate buffer (Figure 2a). At the initial stages of thermal denaturation at 4 °C, the ellipticity was observed to have a sigmoidal dependence on temperature. Then, at 30 °C, the ellipticity remained constant until the temperature reached 60 °C and further increased; again, a sigmoidal-like unfolding transition could be observed. We attributed both sigmoidal processes to cold and hot denaturation (see labels *T_trs, cold_* and *T_trs, hot_* in Figure 2a). Hence, at ambient temperatures (ca. 20 °C), the thermophilic multidomain protein IF2 was largely unfolded (ca. 50%). To obtain the transition midpoint temperatures in a model-independent way, we applied an approach based on the first derivative of the experimental curves. Upon derivation, two extrema were observed: a minimum, which corresponds to the midpoint of cold denaturation; and a maximum, which corresponds to the midpoint of hot denaturation. Both denaturing transitions were cooperative, and the temperatures corresponding to cold and hot denaturation were *T_trs, cold_* 18 ± 0.4 °C and *T_trs, hot_* 67.4 ± 0.2 °C. 

One of the possible mechanisms underlying the thermal adaptation of thermophilic proteins is based on the optimization of electrostatic interactions on the protein surface. The role of electrostatic interactions in determining protein stability can be studied by examining the dependence of protein stability on the concentration different salt types. To understand the role of electrostatic interactions in the stability of thermophilic IF2, we first conducted experiments using selected salts at a very high ionic strength: 1.5 M (Figure 2b). 

Figure 2b shows the thermal denaturation of IF2 in the presence of 1.5 M NaCl, NH_4_Cl, and NaClO_4_, plotted as unfolded fraction (*f*_u_) vs. temperature (*T*). This combination allows us to compare the role of cations of different ranks in the Hofmeister series (Na^+^ versus NH_4_^+^, both chlorides) and different anions (Cl^−^ versus ClO_4_^−^, Na^+^). In the presence of NH_4_Cl, only minor shifts in *T_hot_* toward higher temperatures at higher salt concentrations and a slight shift in *T_cold_* to lower values were observed. In the presence of NaCl, we found that thermal denaturation on the hot side shifted toward higher temperatures at higher salt concentrations. Interestingly, at 1.5 M NaCl, cold denaturation of IF2 was not observed due to a significant shift in the threshold toward lower temperatures that were not achievable under the conditions of the study. The difference between Na^+^ and NH_4_^+^ chlorides indicates that the nature of the cation and its rank in the Hofmeister series affect the midpoint of cold denaturation.

NH_4_^+^ is a chaotropic cation, while sodium is a mild kosmotropic cation that can stabilize the native conformation of IF2. Such stabilization can be offset by a chaotropic anion. Therefore, in the following unfolding experiment, we used a combination of sodium and chaotropic anion-perchlorate. In the presence of 1.5 M NaClO_4_, cold denaturation was again observed with *T_cold_* = 11.6 ± 0.2 °C. The temperature for thermal denaturation on the hot side decreased as expected for chaotropic anion, with *T_trs, hot_* = 63.6 ± 0.4 °C. One can conclude that cold and hot denaturation processes are dependent on the salt type and ionic strength.

We systematically examined the thermal denaturation of IF2 in NH_4_Cl, NaCl, CsCl, LiCl, and NaClO_4_ in the concentration range of 0.1–2 M (Figure 2c gray *T_trs, cold_* and black *T_trs, hot_*). In the presence of NH_4_Cl (Figure 2c), the thermal transition temperature *T_trs, hot_* showed a hyperbolic increase. On the other hand, *T_trs, cold_* displayed a linear decrease, indicating that IF2 becomes more stable and, hence, more folded at ambient temperature. Here, even though NH_4_^+^ is a chaotropic cation, higher transition midpoint temperatures resulted in the stabilization of the protein at elevated temperatures. 

A similar increase in thermal denaturation on the hot side is visible in the presence of NaCl. Here, *T_trs, hot_* increased dramatically; *T_trs, hot_* increased by ca. 15 °C in the presence of 1 M salts and by 20 ^°^C in 2 M NaCl. In the case of cold denaturation, *T_trs, cold_* decreased to a small extent, by 3.8 °C, with 1 M NaCl, as can be seen from Figure 2c. However, cold denaturation was not observed because the CD signal was approximately halved compared with at 0.1 M NaCl. The possible reasons for this are discussed later in this paper. 

Thermal denaturation in the presence of CsCl mirrors the dependence found for NaCl (Figure 2c). Increasing the CsCl concentration significantly increased the transition temperature. At the same time, the threshold for cold denaturation shifted toward lower temperatures, and the cold denaturation process was not visible anymore above 0.5 M CsCl.

LiCl and NaClO_4_ salts were found to share similarities in terms of the concentration dependences of *T_trs, hot_* and *T_trs, cold_*. Here, *T_trs, hot_* decreased by 1.0 ± 0.2 °C (2 M LiCl) or 5.2 ± 0.2 °C (2 M NaClO_4_), which indicates a mild destabilization of native IF2 on the hot side. On the other hand, *T_trs, cold_* decreased by 8.0 ± 0.4 °C (at 2 M LiCl) and 9.8 ± 0.2 °C (at 1.5 M NaClO_4_), indicating IF2 becomes more stable at ambient temperatures.

### 2.2. Limitations to the Analysis of Thermal Denaturation of Two Overlapping Independent Fragments

*Thermus thermophilus* IF2 consists of two parts that can be obtained after proteolytical cleavage [31]: a G fragment and a C fragment. Although the fragments have similar stabilities, two different denaturation pathways are represented. On the other hand, when the denaturation of domains is coupled, it can be described as a two-state process. For fragments displaying similar signals (as in our case, Figure 3a, similar secondary structure content) and independent denaturation, analysis of the thermal denaturation profiles can be misleading. We previously showed [28] that both fragments display similar thermal denaturation profiles; in the presence of 9 M urea, *T_trs, hot_* for the G fragment is 67.6 °C and *T_trs, hot_* for the C fragment is 71.6 °C. Hence, we assumed there are two denaturation pathways (Figure 3b). In one pathway, the denaturation starts from the C fragment, and in the second pathway, the unfolding starts from the G fragment. The observed denaturation process would correspond to the average signal, and the enthalpy and heat capacity would be much lower compared with the coupled all-or-nothing denaturation, where enthalpies and heat capacities add up. We conducted a thermodynamic analysis of IF2 in the presence of 0.1 M NaCl and found values that were too low, as was expected for such a large multidomain protein. To highlight the observed inconsistencies, the thermal profile of the CD signal of IF2 in 0.1 M NaCl was fitted by the Gibbs–Helmholtz equation with the parameters Δ*c_p_* 7.9 kJ/mol/K, Δ*H* 212 kJ/mol, and *T_trs, hot_* 69.1 °C, i.e., with a heat capacity value that is significantly smaller than expected for a 563 aa protein (7.9 vs. expected 33 kJ/K/mol) [11]. We assumed that the experimental curve could be decomposed according to independent unfolding of the G and C fragments, in which transitions are described by the same thermodynamic parameters (Figure 3c). Hence, the observed thermal profile reflects the sum of the unfolding of the independent fragments, with limited possibility to resolve individual fragment unfolding profiles and their corresponding thermodynamic parameters (Figure 3d,e). 

Mathematical Modeling—Effect of Individual Parameters and Transition Temperature (Figure 3f): We examined how the thermal profiles of independently unfolding fragments can be resolved and how they affect the apparent two-state character of thermal curves. The denaturation parameters for the C fragment were held constant (Δ*H* 212 kJ/mol/K, Δ*c_p_* 7.9 kJ/mol, and *T_trs, hot_* 342.2 K) while the parameters for the G fragment were changed as indicated in Figure 3f (variable *T_trs, hot_*, Δ*H* 212 kJ/mol/K, Δ*c_p_* 7.9 kJ/mol). A difference of 10 °C in *T_trs, hot_* between fragments did not allow resolving individual unfolding transitions. Under the parameters of the study, a difference of at least 20 °C was needed to resolve transitions in both the hot and cold sides. From a technical viewpoint, multistate behavior on the hot side was more evident due to the presence of post- and pretransition baselines. This situation was reversed when *T_trs, hot_* shifted by at least 20 °C toward a higher temperature; here, the cold denaturation profile exhibited clear post- and pretransition baselines and, hence, fragment unfolding became visible. To conclude, even with a 15 °C difference between *T_trs, hot_* of the fragments was not sufficient to allow distinguishing between individual transitions. 

Enthalpy (Figure 3g): A decrease in the enthalpy of the G fragment would result in an evident fingerprint of multistate unfolding. Importantly, the simulated enthalpies (112 and 12 kJ/mol) were quite low and on the small side of the usual enthalpy range. Moreover, post- and pretransition baselines were not reached due to the low stability of the native form; under any conditions, the native state is not 100% present, which may bias the analysis [32]. The increasing denaturation enthalpy of the G fragment and high Van ’t Hoff enthalpies do not allow for a clear resolution between the individual denaturation curves. 

Heat capacity change (Figure 3h): In the last simulation set, we changed the values of the heat capacity of the G fragment while keeping all other parameters constant, including the parameters for the C fragment (Figure 3h). A decrease in the heat capacity from 5.9 to 2.9 kJ/mol/K leads to changes in the cold denaturation profile; however, the changes in the profiles are difficult to analyze using the three-state model. The thermal denaturation was not affected at high temperatures because, *T_trs, hot_* on the hot side was taken as constant in our simulation. An increase in the unfolding heat capacity from 12.9 to 17.9 kJ/mol/K results in a very clear thermal profile with visible multistep unfolding processes. However, at very high values of Δ*C_p_*_,_ the native baseline is not reached (i.e., a temperature range where the native state is population is effectively 100 % [32]), which hampers the analysis of the midpoints. 

In our initial effort to analyze the thermal denaturation of IF2, we used the first derivative of the experimental curve. In such an approach, issues associated with the resolvability of thermal denaturation transitions may be encountered. Based on the same data spacing used in our experiments, we compared outcomes from the first derivative with the expected values for the melting temperatures (see Appendix A, Appendix A). We found that the temperature difference between the melting temperatures of the fragments must be at least ±20 °C to detect the second melting temperature. The transition temperatures of the fragments did not change in our simulations; hence, the unfolding transitions differed only due to changes in the enthalpy change or heat capacity, and we did not observe the second melting temperature on the hot side of the thermal denaturation curve. Only on the cold side and under a special set of conditions were two cold denaturation steps visible, as highlighted by the presence of two minima of the derived curve. 

In summary, simulations of independent thermal denaturation transitions were useful for achieving resolution based on our experimental method. Our simulations indicate the highly limited resolution of the denaturation curves and thermodynamic parameters for the full-length protein because of the significant overlap in the denaturation curves for both fragments. Hence, instead of studying the full-length protein, we focused on the individual fragments: the G and C fragments.

### 2.3. Salt-Dependent Thermal Denaturation of Individual Fragments Reveals Differences in Salt-Screenable Electrostatic Interactions

As indicated by our simulations, it is important to dissect multidomain proteins and separately analyze their fragments. To this end, we prepared and isolated individual fragments of IF2 (see [28]). Thermal denaturation using circular dichroism in the far-UV region showed that both fragments undergo cold and hot denaturation (Figure 3a). Using nonlinear regression analysis of the two-state model, we were able to obtain the thermodynamic values Δ*H,* Δ*c_p_*, and *T_trs, hot/cold_* for the individual transitions. Due to cold and hot denaturation, all parameters could be obtained with reasonable accuracy. We found that Δ*H_app_* and Δ*c_p, app_* of the individual fragments are similar regarding their full-length IF2. We wondered whether the stability of individual fragments would also display the same salt and ionic strength concentration dependence. To this end, the thermal denaturation experiments were conducted for both fragments and analyzed. The thermodynamic data are shown in Figure 4a–d ac (G fragment in red and C fragment in blue) and are in agreement with the values found for other proteins [11]. 

In the presence of NH_4_Cl, the unfolding enthalpy for both G and C fragments decreased. However, the salt dependence of the G fragment, was slightly higher. The same was true for Δ*c_p, app_*, which decreases significantly at high NH_4_Cl concentrations, from 8.9 to 4.5 kJ/mol/K at 2 M. Again, the Δ*c_p, app_* was slightly more salt-sensitive for the G fragment. In 0.1 M NH_4_Cl, the transition temperature was 65.0 °C for the G fragment and 70.0 °C for the C fragment. As part of the full-length IF2, the *T_trs, hot_* fragments were separated by less than 5 °C; hence, we could not resolve the individual transitions. *T_trs, hot_* for both fragments were dependent on NH_4_Cl, and we observed that *T_trs, hot_* was more salt-dependent for the C fragment than the G fragment. 

A much larger difference and high salt sensitivity were found for NaCl; again, the unfolding enthalpy and heat capacity change decreased at high salt concentrations of NaCl. This change was even more pronounced for the G fragment when Δ*H_app_* reduced from 212 to 50 kJ/mol at 2 M NaCl. At the same time, the C fragment Δ*H_app_* decreased from 212 kJ/mol at 0.1 M NaCl to 138.5 kJ/mol at 2 M NaCl. The salt-induced change in heat capacity is clearly visible for both fragments, with Δ*c_p, app_* reduced to 4 kJ/mol/K for the C fragment and 2 kJ/mol for the G fragment. Again, the slope is higher for the G fragment, indicating higher salt sensitivity. The transition temperature in 0.1 M NaCl is 65.7 °C for the G fragment and 71.5 °C for the C fragment. The difference between *T_trs, hot_* for both fragments increases dramatically due to a significant drop in *T_trs, hot_* of the G fragment at a high salt concentration, highlighting the high salt sensitivity of the G fragment. At 2 M NaCl, *T_trs, hot_* is 56.4 °C for the G fragment and 85.1 °C for the C fragment, which are substantially different and could, therefore, be potentially observable in full-length IF2. Our mathematical simulations (see Section 2.2) were performed with one variable parameter (e.g., *T_trs, hot_*), while the other thermodynamic parameters remained fixed (e.g., Δ*H*, Δ*c_p_*). Our analysis of experimental data clearly shows that all parameters are salt-dependent. Hence, they may compromise the ability to resolve individual transitions even if the difference in *T_trs, hot_* is significant enough for a certain combination of the thermodynamic parameters. Using experimentally obtained thermodynamic parameters for individual domains at 2 M NaCl, we calculated the profiles of the individual denaturation curves of G and C fragments. Again, we used temperature steps and extracted the noise from experiments. After the simple addition of the calculated denaturation curves of fragments, we obtained a hypothetical denaturation curve for full IF2. Then, the calculated full IF2 denaturation curve was analyzed and subjected to visual assessment. We found that even a 28 °C difference in transition temperatures for the G and C fragments is insufficient when the denaturation enthalpies and heat capacities are very low, as was observed with 2 M NaCl. In summary, in the presence of NaCl and CsCl, the G fragment displays nonlinear salt dependence of transition temperatures, while the C fragment displays linear dependences. Individual denaturation processes for G/C domains could not be observed in the full-length protein because the small magnitude of enthalpy and heat capacity changes further decreased the resolution. 

We found that in the presence of CsCl, the differences between G and C fragments were often very subtle and with similar values for Δ*H_app_* and Δ*c_p, app_*. Decreases in Δ*H_app_* were observed at high salt concentration, with only a slight difference in the salt dependence slope for G and C fragments. As was found for other salts, Δ*c_p, app_* decreases as the salt concentration is increased. Here, Δ*c_p, app_* for G and C fragments demonstrated nearly equal salt sensitivity and slope compared with variations in salt. In 0.1 M CsCl, the transition temperature was 65.5 °C for the G fragment and 71.1 °C for the C fragment. Similarly, as for NaCl dependence, the difference between *T_trs, hot_* for both fragments increased dramatically due to the significant nonlinear drop in *T_trs, hot_* of the G fragment at high salt concentration, hence highlighting the high salt sensitivity of the G fragment.

At 2 M CsCl, *T_trs, hot_* was 50.5 °C for the G fragment and 79.0 °C for the C fragment, which represents a substantial difference that could be potentially observable in full-length IF2. Again, as in the case of 2 M NaCl, we simulated the individual unfolding transitions for the G and C fragments. Here, similar to our previous observations, the resolution was compromised due to a large decrease in enthalpy and heat capacity. Hence, we were not able to resolve different denaturation curves for the fragments.

In summary, the G fragment displayed a higher and nonlinear salt dependence than the C fragment. In 0.1 M NaClO_4_, the transition temperature was 58.1 °C for the G fragment and 68.6 °C for the C fragment. As a part of the full-length IF2, the *T_trs, hot_* of fragments differed by less than 10.5 °C; hence, we could not resolve the individual transitions. *T_trs, hot_* for both fragments decreased at high perchlorate concentrations. Here, the G fragment showed less salt sensitivity than the C fragment. The difference between *T_trs, hot_* decreased, and G and C fragments had similar melting temperatures at 1 M NaClO_4_, causing them to be indistinguishable in thermal denaturation experiments of the full-length protein. At higher concentrations, the melting temperatures differed only slightly for both fragments.

### 2.4. Salt-Dependent Thermodynamic Parameters for Denaturation of the G and C Fragments 

From the dependence of thermodynamic parameters on salt concentrations, we quantified how strongly these parameters depend on salt. In particular, we defined the slope of the salt dependencies (e.g., ΔΔ*H*/Δ[salt]) as a suitable measure to determine the extent to which salt influences the denaturation process. By comparing different salt types, one can potentially learn about the molecular mechanism behind the salt-specific effects on cold denaturation (Figure 5a–c). 

Salt-specific enthalpy change: Overall, the G domain showed much higher sensitivity to salts than the C fragment. In absolute values, the G fragment had a higher slope (red versus blue in Figure 5a). Under all conditions, the slopes were negative and dependent on the salt type. The largest negative slope was observed for CsCl, while NH_4_Cl and NaClO_4_ had the lowest slope values. The profile of salt-specific slopes was similar for G and C fragments. In summary, the analysis showed that the change in denaturation enthalpy was dependent on the type of cation and anion, and denaturation enthalpy was more highly dependent on salts for the G fragment than the C fragment. 

Salt-specific heat capacity change: Again, the concentration dependence of the heat capacity was found to be higher for the G domain than the C fragment. The only exception is CsCl, where both fragments had similar slopes. The highest negative slope was observed for NaCl and the G fragment, and the lowest negative slope for LiCl and the C fragment in the case of chloride ions. The slope became slightly positive in the presence of NaClO_4_ for the heat capacity of the C fragment, while the G domain heat capacity had a significantly negative slope. As previously observed, the slopes were dependent on the type of salt, and the profiles observed for the G and C fragments were similar. In addition, the C fragment heat capacity showed high anion dependence when comparing the slopes of sodium chloride with those of sodium perchloride anion. The change in the slopes was much larger compared with that observed for the G fragment. 

Salt-specific transition temperature changes *T_trs, hot_*: Transition temperatures have only been discussed on the hot side, where we have enough data points at different salt concentrations. For the cold transition, we were not able to reliably estimate the slope due to technical limitations. The G fragment transition temperatures showed high positive slopes for CsCl, while the C fragment transition temperatures showed the highest slope in NaCl. The profiles of the slopes for different salts were similar, even though differences in magnitude were present. In the case of perchlorate, both slopes were negative; however, the C fragment transition temperatures had a larger negative slope than those of the G fragment. 

## 3. Discussion

### 3.1. Thermophilic Multidomain IF2 Undergoes Cold Denaturation

Thermophilic proteins are usually heat-stable proteins and, hence, attractive for industrial applications. Is a high melting temperature always a good indicator of stability at room temperature? Here, we report that this is not always the case due to the cold-denatured state of proteins, which is little-understood compared with the heat-denatured state. A cold-denatured state may also have some physiological relevance, as indicated by the recent findings of Horowitz et al. [36]. Interestingly, the ordering of water molecules around the unfolded protein substrates in the GroEL/ES chaperonin cage resembles a cold-denatured state [36]. Compared with thermal heat denaturation studies, cold denaturation studies are more sparse because they can only be observed under specific experimental conditions. By employing large reversibly denatured proteins from thermophilic bacteria, we found conditions where the protein undergoes cold denaturation. It should be noted that in the absence of destabilizing agents, no cold denaturation was observed. However, for industrial applications, proteins and enzymes are often used in the presence of organic solvents, at nonphysiological pHs, and ionic liquids. These can strongly destabilize the protein and, hence, may resemble the conditions used in our experiments. In our case, cold denaturation of IF2 was observed in the presence of high urea concentrations and also in guanidinium chloride. High urea concentrations were initially used to suppress aggregation of the C domain. We observed in one of our experiments that the protein was largely unfolded at ambient temperatures and refolded at higher temperatures. This effect was later confirmed for IF2 and for the G fragment. Urea enabled us to examine the molecular properties of cold denaturation, which is not observed in the absence of urea. Hence, we were able to evaluate the role of salt-screenable electrostatic interactions with different types of salts. Such examination would be much not possible if ionic denaturants, e.g., guanidinium chloride or thiocyanate, were used. Next, even in the presence of 9 M urea, the IF2 protein denatures at a relatively high melting temperature of 67 °C, and the protein remain reasonably stable. The major disadvantage and limitations of this method are the high nonphysiological concentrations of urea, its effect on water structure and activity, and the fact that urea can decompose and chemically react with proteins, leading to the formation of various chemical modifications in these proteins. For example, the decomposition of urea at high temperatures can generate isocyanic acid, which can react with the amino groups of lysine residues in proteins, forming carbamylated lysine residues [37]. Nevertheless, we observed high repeatability of thermal and cold denaturation. 

The high melting temperature of *Thermus* IF2 in the presence of urea indicates high resistance against chemical denaturants and heat resistance. However, at room temperature, a large amount of the protein is unfolded as the protein undergoes cold denaturation. Therefore, a significantly high melting temperature does not guarantee high native state stability at room temperatures. Our study highlights the importance of studying denaturation processes at the lowest possible temperature, at approximately 2–5 °C in our case. 

In our study, we first examined how a salt screening of the electrostatic interactions affects cold denaturation. We found that three selected salts (Figure 2b; NaCl, NH_4_Cl, and NaClO_4_) displayed specific effects on cold denaturation. Clearly, sodium chloride was very effective in suppressing cold denaturation, and we found that the other two salts were much less effective. To find out whether the suppression of the cold denaturation followed the Hofmeister series, we studied the effect of different salts (Figure 2c). In the presence of any of the salts, the transition temperature of the cold denaturation decreased. In contrast, the transition temperature for the hot denaturation could increase/decrease or remain constant. This asymmetric effect of salts on transition temperatures indicates differences in the molecular mechanism of the temperature-dependent protein-solvent interactions at high or low temperatures. Salts display different stabilization mechanisms. Sodium and cesium cations stabilize the native state at elevated temperatures, as indicated by the increase in melting temperature at high temperatures and via the strong suppression of the cold denaturation. Other salts, such as NH_4_Cl, LiCl, and chaotropic NaClO_4_ (Figure 2c, black dots) have a modest, nonexistent, or even a decreasing effect on *T_trs, hot_*. However, this destabilization is not seen in the cold denaturation process, and melting temperature always decreases to different extents depending on the salt type. Hence, these salts cause water-hydrophobic interactions to be even less favorable, which leads to destabilization of the cold denatured state and decreased T*_trs, cold_*. We could not achieve cold denaturation at higher temperatures under any tested set of conditions. 

### 3.2. Thermodynamic Analysis of Unfolding Denaturation of Multidomain Proteins

As we previously found, IF2 denaturation is a reversible process [28], which makes it possible to apply thermodynamic analysis. The presence of cold and hot denaturation processes enables the resolution of a complete protein stability curve and the evaluation of the heat capacity change. The analysis of the hot denaturation process is only conducted over a very narrow temperature range that is insufficient to allow a reliable estimation of the heat capacity change. Therefore, during the analysis of hot denaturation, this change is often assumed to be zero or some other value obtained in other investigations. Using a traditional approach, a Van ’t Hoff plot is constructed from experimental data. For this plot, only a limited range of temperatures is used—usually 5–95% of the progress of the denaturation. In our preliminary analysis, we found that Van ’t Hoff plots are well approximated by the linear function, which indicates that the enthalpy does not depend on the temperature within the limited range of temperatures. Hence, the major advantage of nonlinear regression analysis of both cold and hot denaturation process is that the curvature of the protein stability curve is quite reliable. 

The heat capacity change is directly dependent on the number of residues involved in the transition, which is proportional to the change in the solvent-accessible area. The solvent-accessible area can be dissected into polar and nonpolar areas, for which the heat capacity change has opposite signs. Janin found that folded proteins bury a constant fraction of the polar surface area [38], which makes it difficult to intersect individual contributions of polar and hydrophobic surfaces. This is indirectly supported by another analysis [11,39], resulting in a correlation between the number of residues and heat capacity change observed in the seminal Robertson and Murphy review [11]. Based on the analysis of the folded proteins, one can assume the size of the heat capacity change for a cooperative unit with an IF2 of a specific size. For a protein with the size of 571 aa, the predicted heat capacity change should be approximately 33 kJ/mol/K, assuming a fully cooperative all-or-nothing system in which the denaturation of domains is fully coupled. This number contrasts with the experimentally found 12 kJ/K/mol value and indicates that the measured cooperative unit is much smaller than for the fully folded protein. At the same time, the denaturation enthalpy at 60 °C [11] is also correlated with the number of residues; at a size of 571 aa, and the predicted value for Δ*H* is 1440 kJ/mol. The predicted value contrasts with the experimentally found ca. 200 kJ/mol value, which is seven-fold lower. The more considerable seven-fold difference in the enthalpy, compared with the three-fold difference in heat capacity, could be due to the enthalpic contribution of the urea–protein interaction. To conclude, the thermodynamic analysis of IF2 indicates a much smaller cooperative unit undergoing denaturation. Multidomain IF2 is well-folded and stable [28]; therefore, we can rule out that some large parts of the protein are intrinsically disordered, which would also result in a low heat capacity and enthalpy of the denaturation.

We have previously shown that IF2 consists of autonomous and largely independent folding units with similar thermodynamic properties [28]. If the denaturation of the fragments proceeds in an uncoupled, independent manner, the observed thermal denaturation appears as an apparent two-state reaction when the stability of the fragments is similar. Indeed, in analyzing the hot and cold denaturation of the individual fragments, the transitions occur within a similar temperature range. In other reported cases, such as the denaturation of gene-3-protein, multidomain and multistage unfolding is clearly visible, enabling accurate thermodynamic parameter assessment [40]. In the case of IF2, a significant overlap in the melting transitions leads to the averaging of the melting curves. We examined the conditions when independent denaturation transitions merged, and they could not be resolved using our analysis. Using mathematical modeling of the independent transitions that systematically varied values for *T_trs, hot_*, Δ*H*, and Δ*c_p_*, we could decipher the resolution of our experimental methods. To this end, it is crucial to mimic experimental conditions as much as possible, which include finite temperature steps and noise. The noise was extracted from an experimental curve, and this noise is, to a large extent, temperature independent. After calculating the first derivative of the signal for temperature, we found that even large differences in Δ*H*, Δ*c_p_*, and *T_trs, hot_* may not be enough to allow observing the distinct multiple steps of the denaturation process, particularly when denaturation enthalpies are low. Hence, under our conditions, even a 20 °C theoretical difference in *T_trs, hot_* between unfolding G and C fragments would not result in a separate denaturation curves. When enthalpy decreases even further in the presence of salts, the resolution might be even worse. For example, in 2 M NaCl and CsCl, *T_trs, hot_* of the C and G fragments differed by 20 °C; however, the transitions were not resolved in either the simulation or experiments. This is mostly due to the decrease in denaturation enthalpy (50 kJ/mol for G fragment and 212 kJ/mol for C fragment in the presence of NaCl; and 36 kJ/mol for G fragment and 169 kJ/mol for C fragment in the presence CsCl). As such, care must be given to the analysis when the enthalpies are low, because the analysis can be corrupted. This is because the native state is not fully reached, which affects the signal for the native state and native state baselines [32]. 

In the abovementioned approach, we utilize the first derivative of the signal concerning temperature, which has a clear limit due to the temperature increment over which the signal was measured, and also due to the presence of noise that had to be eliminated by averaging or by carefully performed smoothing. The major advantage of the first derivative approach is that it is model-independent and can, in principle, resolve individual transitions. Here, one interesting observation is that cold denaturation may provide a better resolution and identification of the multiple denaturation processes than hot denaturation. For example, we detected two transitions in the cold denaturation process, while during hot denaturation, there was only a single denaturation process with no transition. In the case of phosphoglycerate kinase, the authors in [41] observed two transitions during cold denaturation, while they observed only one denaturation transition on the hot side. One can conclude that the denaturation process involves a different number of steps and states on the cold and hot sides. Our analysis suggests that the number of observable transitions is not a secure diagnostic tool to conclude differences in the denaturation processes, i.e., the process can involve the same number of steps that are not resolvable due to the overlap in the transitions. Importantly, several prerequisites have to be fulfilled to reach robust conclusions on the thermodynamics of the denaturation process. 

First, the protein undergoes reversible denaturation IF2 and the fragments undergo reversible denaturation, as evidenced by the repeatability of the observed cold and hot denaturation (4–100 °C). Second, for valid two-state analysis, the cold- and heat-denatured states must be thermodynamically equivalent, which enables a description of cold and hot denaturation by a single temperature-dependent equilibrium constant. If not, these processes must be described by two distinct equilibrium constants. In our case, we observed the equal intensity of the CD signal at 222 nm for the cold- and heat-denatured states. As a CD signal at 222 nm is a highly sensitive probe for the secondary structure of a protein, identical values for the cold- and heat-denatured states indicates that these states did not differ in terms of their secondary structure content. Moreover, they are structurally equivalent and, therefore, also very likely to be thermodynamically equivalent. 

To summarize, the thermodynamic values for multidomain proteins obtained from spectroscopic methods must be observed cautiously, since they represent an averaged behavior when domains independently denature and the transitions overlap with less than a 20 °C separation.

### 3.3. Salt-Screenable Electrostatics of Interdomain Interactions

We observed an apparent decrease in the denaturation enthalpy in the presence of all salts (Figure 5a, negative slopes for ΔΔ*H/*Δ*[salt]*). A negative slope indicates that electrostatic interactions in the native state actually contribute favorably, in a highly enthalpic manner, to the stability of IF2. Furthermore, the presence of salt screens out favorable interactions. As mentioned above, the decrease in enthalpy is also accompanied by the loss of heat capacity change, which can be due to changes on the native side, for example, in the size of the cooperative units or changes in the compactness of the unfolded state. Based on the analysis of the CD profiles of thermal transition, we can exclude the formation of an even more profound residual structure in the unfolded state. Therefore, the loss of enthalpy and heat capacity change is likely due to a decrease in the cooperative unit. A plausible explanation for this is the loss of folding cooperativity between the individual domains in the G and C fragments. In the case of the G fragment, the G1 domain is clearly separated from the other G2/G3 domains; therefore, G1 can be seen as a candidate for an autonomous, thermodynamically stable, folded unit. For the C fragment, we can identify two C1/C2 domains, which are also separated in space; therefore, it is plausible that the domains can independently denature under some conditions, e.g., in the presence of the salts. We observed that the changes in denaturation enthalpy and heat capacity are more salt-sensitive (higher absolute slope values) in the domains of the G fragment than for the C fragment. This difference may arise from the different number of domains in the G and C fragments, and from inherent differences in the surface electrostatics.

Interestingly, with the exception of perchlorate, most salts increase the melting temperature, indicating stabilization due to salts is entropic. A favorable entropic term may be attributed to the native state increase in the entropy upon losing the domain–domain interactions. As opposed to the C domains, G domains consist of several clusters of charged residues (Figure 5). Again, a higher number of positively charged residues may also result in higher salt sensitivity and, hence, higher absolute values of the slopes of Δ*H* vs. salt concentration. In the case of perchlorate, the native state is destabilized, as indicated by decreases in both melting temperature and denaturation enthalpy. However, cold denaturation is not shifted toward higher temperatures, which might be expected in the case of the native state destabilization relative to the unfolded state. The change in the heat capacity of the C fragment is slightly positive, in contrast to the G fragment, where perchlorate dramatically decreases heat capacity. The type of anions appears to have a much greater influence when one compares sodium chloride with perchlorate. This might be due to the screening efficiency of the localized exposed positive charge in the C fragment (Figure 5d).

The observed slopes of thermodynamic parameters obtained for different salts (e.g., ΔΔH/Δ[salt]) are specific for a given chloride salt and may be correlated with some of the physicochemical properties of the cation. Similar correlations were found in different studies of proteins in the presence of salts of the Hofmeister series [42,43,44,45]. In our case, we could not find any evidence for a robust correlation between the salt-specific values of the slopes and ion properties such as hydration entropy, hydration number, charge density, polarizability, and viscosity B coefficient (see Appendix A, Appendix A, which includes also pairwise correlations between thermodynamic parameters of domains). The lack of a clear correlation can be possibly attributed to the complexity of the effects of cations on stability and resolvable transitions and the multistate denaturation process, which we addressed in this work.

**Figure 5 ijms-24-06787-f005:**
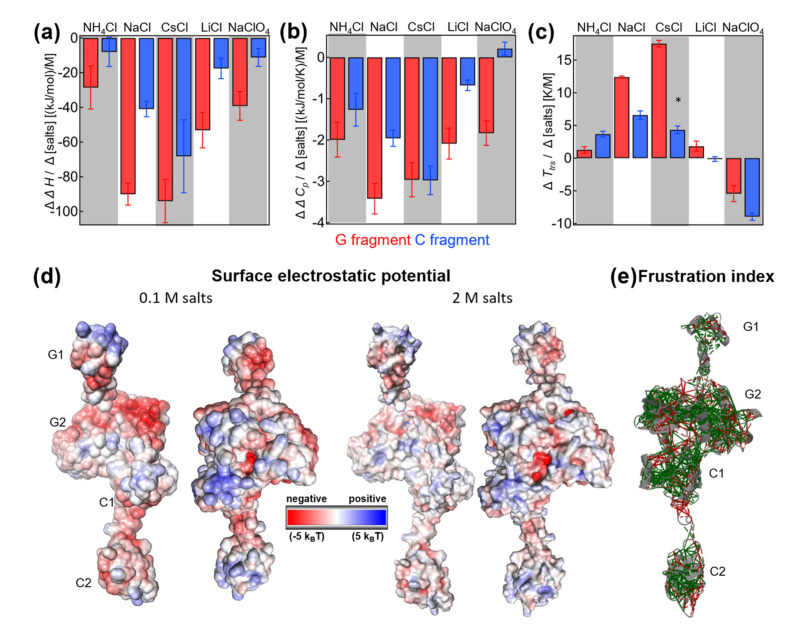
The salt-concentration-dependent changes in enthalpy (**a**), heat capacity (**b**), and temperature of transition (**c**) for the G fragment (red) and C fragment (blue). (**d**) Distribution of surface electrostatic potential in the presence of 0.1 M salt (left) and 2 M salt (right). (**e**) The frustration index analysis performed by [43,44,45]. The color of the lines indicates whether a given contact is minimally (green) or highly (red) frustrated. The energy of a contact between two amino acids in a protein determines how favorable that interaction is. A contact is considered “minimally frustrated” if its energy is on the lower end of the possible range of energies for that type of contact. Conversely, a contact is considered “highly frustrated” if its energy is on the higher end of the range. For more details, please see the Discussion and Materials and Methods sections. Significant correlations and linear dependencies (≤5% probability, yellow background in Appendix A) are shown as stars (*).

To analyze the internal organization of the domains of IF2, we performed frustration analysis [46,47,48]. The server that uses the so-called frustratometer algorithm classifies individual contacts based on their frustration index value. If a contact has a high frustration index it is relatively unfavorable compared with other possible contacts in that location. Conversely, if it has a low frustration index, it is relatively favorable compared with other possible contacts.

Contacts with a frustration index that falls within a certain range are classified as “minimally frustrated”, “highly frustrated”, or “neutral.” A contact is considered “minimally frustrated” if its native energy is at the lower end of the distribution of decoy energies, indicating that most other amino acid pairs in that position would be unfavorable. A contact is considered “highly frustrated” if its native energy is at the other end of the distribution of decoy energies, indicating that most other amino acid pairs at that location would be more favorable for folding than the native ones. Contacts that fall in between these limits are classified as “neutral.” Highly frustrated contacts can be problematic for protein stability and function because they may lead to local unfolding or misfolding of the protein. Identifying highly frustrated contacts can assist in understanding the structural and functional properties of a protein and can guide protein engineering efforts to improve stability and function. The analysis highlights that, while intradomain interactions show minimal frustration, the interdomain interaction between G domains, in particular, is highly frustrated (Figure 5e). Such a large frustration of interdomain interactions may have a biological significance due to the need for structural adaptation after binding to the ribosome. Upon binding, IF2 has to satisfy many contacts with other interacting sites on the ribosome and other molecules such as tRNA. The structurally encoded loss of the interdomain cooperativity may provide a suitable strategy to control the protein conformation by weakly stabilizing the frustrated internal network of electrostatic interactions. This control enables the protein to be compact in the absence of a ribosome. At the same time, upon binding and electrostatic shielding, the domains can now work more independently, which also allows securing contacts at distal sites.

## 4. Materials and Methods

### 4.1. Chemicals

All chemicals were obtained from Sigma-Aldrich: sodium cacodylate trihydrate (Product No. C4945); sodium phosphate dibasic (Product No. 71642); sodium phosphate monobasic (Product No. 71496); urea (Product No. U5128); ammonium chloride (Product No. 213330); sodium chloride (Product No. S9888); cesium chloride (Product No. 289329); lithium chloride (Product No. 310468); and sodium perchlorate monohydrate (Product No. 310514). 

### 4.2. Protein Preparation and Purification

As previously described [28], IF2 from *T. thermophilus* was purified according to the protocol of Vornlocher et al. [31]. The C-terminal domain (residues 364–571) (IF2C) and the N-terminal domain (G domains, residues 60–364) (IF2G) of *T. thermophilus* IF2 were initially obtained with the preparative limited proteolysis of the intact protein with trypsin and thermolysin, respectively. The N-terminal segment (1–60) was found to have an unordered structure upon isolation and, thus, was not further analyzed.

Digestion products were separated by chromatography on a Q-Sepharose column (Pharmacia, Uppsala, Sweden). For expression of the C-terminal domain, the coding DNA sequence was amplified via PCR using the following primers: 5′-CAT ATG CAG GAG GAG GGG CGT AAG GAG CTC AAC C-3′ and 5′ g GAG CAT AAG CTT AGG CGG GGA CCT CC- 3′ (an AGG codon was changed into CGT for efficient expression in *Escherichia coli*). For the overproduction of IF2G, the coding DNA sequence was amplified via PCR using the following primers: 5′-CAT ATG GCC AAG GTA CGT ATC-3′ and 5′-GGA TCC GGT CCG GGG GCG GCG-3′ (an AGG codon was changed into CGT for efficient expression in *E. coli*). The resulting DNA fragments were cloned into pET28a (Novagen, Merck, Darmstadt, Germany), and the corresponding polypeptides were expressed with an N-terminal His6 tag in *E. coli BL21(DE3)* (Novagen, Merck, Darmstadt, Germany). The nucleotide sequence was verified with sequencing. His6-tagged IF2C and IF2G were purified with affinity chromatography on the Ni-NTA agarose (Qiagen, Venlo, The Netherlands). The His6 tag was removed through digestion with thrombin, and the purified polypeptides were verified with N-terminal sequencing.

### 4.3. Circular Dichroism and Sample Preparations

All measurements were performed with protein samples dialyzed overnight against 50 mM sodium phosphate (pH 7.0) (IF2 and IF2 g fragment) and 50 mM sodium cacodylate (pH 7.0) (IF2-C fragment). The concentration of proteins was determined with the molar extinction coefficients at 280 nm [49]: ε = 25,900 M^−1^ cm^−1^ (IF2), ε = 9970 M^−1^ cm^−1^ (IF2 g fragment), and ε = 15,930 M^−1^ cm^−1^ (IF2-C fragment). Urea (from Sigma) solutions were freshly prepared in an appropriate 50 mM buffer. Urea concentrations were determined by refractive index measurements (Abbe refractometer, CETI, Liege, Belgium) [50]. Importantly, high-urea solutions can precipitate at the low temperatures used in our experiment. We found that the filtered urea solutions did not precipitate, which was also confirmed by the simultaneous absorbance measurements during the temperature-ramping experiments (see Appendix A, Appendix A).

Circular dichroism (CD) measurements were recorded in 50 mM sodium phosphate pH 7.0 (IF2 and IF2 g fragment) and 50 mM sodium cacodylate pH 7.0 (IF2-C fragment) using a JASCO J-810 (Tokyo, Japan) spectropolarimeter with 5–10 µM protein samples, a 9 M concentration of urea and 0.1–2 M concentration of salts. The thermal transition was monitored by ellipticity at 222 nm at a constant heating rate of 1 °C/min. The temperature of samples was controlled with a Peltier element. The thermal transition measurements in the far-UV region were performed with a bandwidth of 2 nm, response time of 4 s, and standard sensitivity and measurement range of 4–98 °C using a rectangular cuvette with a path length of 1 mm. The measurements started from the unfolded state of 4 °C.

### 4.4. Nonlinear Regression Analysis

The measured data were analyzed and fitted using nonlinear regression analysis implemented in Igor Pro (WaveMetrics, Lake Oswego, OR, USA) using the following equation [51]:(1)$$S=\frac{S_{f}\left(T\right)+\left[S_{u}\left(T\right)\times e^{\left(-\frac{\Delta G\left(T\right)}{RT}\right)}\right]}{1+e^{\left(-\frac{\Delta G\left(T\right)}{RT}\right)}}$$
where *S* is the fitted signal; *S_f_*(*T*) is the measured signal at which the protein is in the folded state as the function of temperature; *S_u_*(*T*) is the measured signal at which the protein is unfolded as the function of temperature; *R* is the gas constant; *T* is the temperature in K; and Δ*G*(*T*) is the Gibbs free energy calculated according to [52]:(2)$$\Delta G\left(T\right)=\Delta H_{T=T_{trs,\;hot}}\left(1-\frac{T}{T_{trs}}\right)+\Delta c_{p}\left[\left(T-T_{trs}\right)-T\;\ln\left(\frac{T}{T_{trs}}\right)\right]$$
where Δ*H_T=Ttrs, hot_* is the change in enthalpy at *T_trs, hot_*; *T_trs, hot_* is the transition temperature; and Δ*c_p_* is heat capacity change.

### 4.5. Theoretical Frustration Analysis of Protein Structures

Frustration analysis [46,47,48]: The frustratometer is an algorithm that uses the energy landscape theory to quantify the degree of local frustration in protein molecules. Frustration is a concept used to understand how energy is distributed in protein structures and how mutations or conformational changes affect energy distributions. High levels of local frustration can indicate important regions such as binding or allosteric sites, while minimally frustrated linkages are indicative of the stably folded core of the molecule. The frustration index is a measure of how favorable a particular contact between two amino acids is relative to all possible contacts in that same location. It is calculated by comparing the energy of the native contact (i.e., the contact found in the protein’s natural folded state) with the energies of many decoy contacts (i.e., other possible contacts that could theoretically exist in that same location). The distribution of energies for these decoy contacts is used to normalize the energy of the native contact and obtain the frustration index. The server that uses the frustratometer algorithm classifies individual contacts based on their frustration index value. If a contact has a high frustration index, it is relatively unfavorable compared with other possible contacts in that location. Conversely, if a contact has a low frustration index, it is relatively favorable compared with other possible contacts. Contacts with a frustration index that falls within a certain range are classified as “minimally frustrated”, “highly frustrated”, or “neutral”. A contact is considered “minimally frustrated” if its native energy is at the lower end of the distribution of decoy energies, indicating that most other amino acid pairs in that position would be unfavorable. A contact is considered “highly frustrated” if its native energy is at the other end of the distribution of decoy energies, indicating that most other amino acid pairs at that location would be more favorable for folding than the native ones. Contacts that fall in between these limits are classified as “neutral.” The thresholds used to define “minimally frustrated” and “highly frustrated” contacts are determined using Z-scores, which measure the number of standard deviations from the native energy that deviate from the mean energy of the decoy contacts. A contact is considered “minimally frustrated” if its Z-score is above a certain threshold and “highly frustrated” if its Z-score is below another threshold.

## Figures and Tables

**Figure 1 ijms-24-06787-f001:**
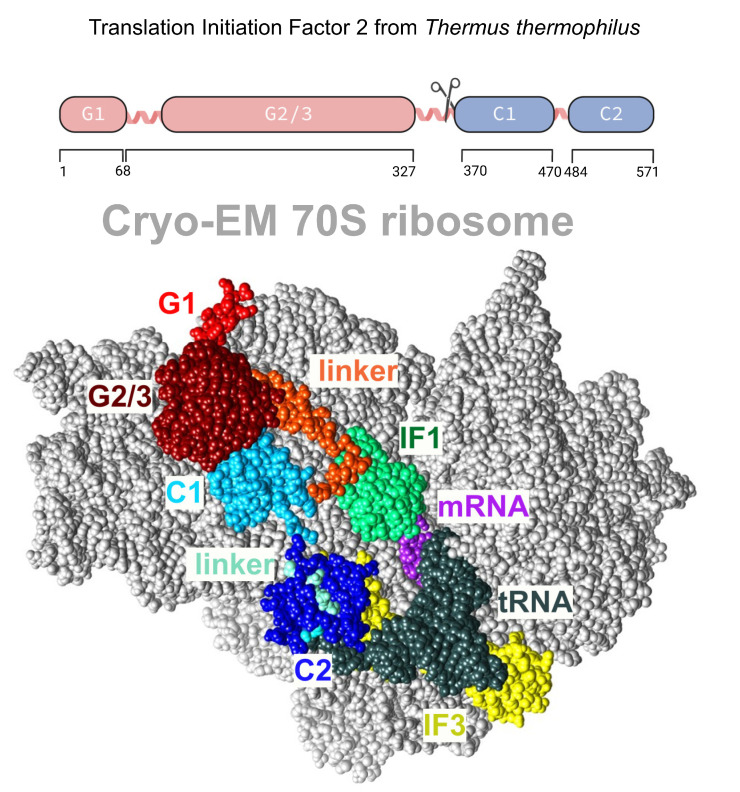
The division (upper part) of translation initiation factor 2 from *Thermus thermophilus* into the G fragment (red), consisting of G1 and G2/3 domains, and the C fragment (blue) consisting of C1 and C2; cryo-EM model (lower part) of ribosome-bound IF2 prepared based on the structure in PDB [5LMV], with binding partners (IF1, tRNA, mRNA, and IF3) shown as colored molecular surfaces.

**Figure 2 ijms-24-06787-f002:**
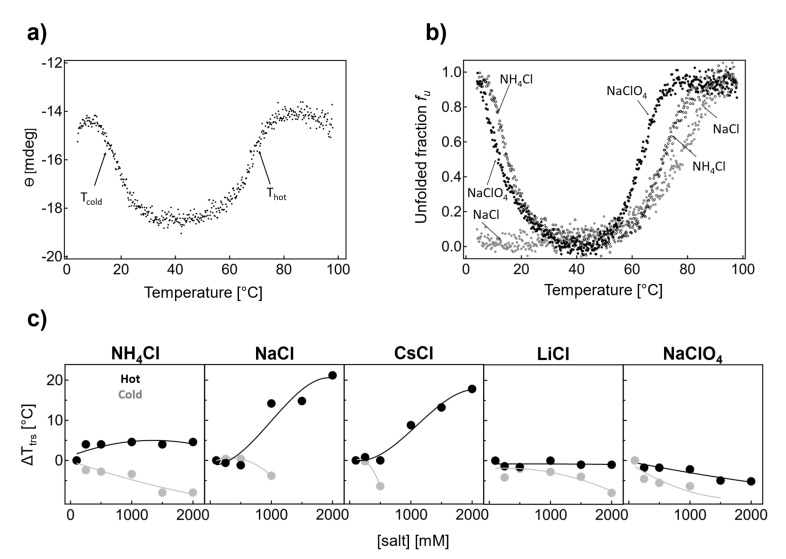
(**a**) Thermal unfolding of IF2 in the presence of 0.1 M NaCl, 9 M urea in the buffer monitored by circular dichroism in the far−UV region at 222 nm. (**b**) Thermal denaturation of IF2 in the presence of 1.5 M NaCl, NH_4_Cl, and NaClO_4_ plotted as an unfolded fraction (f_u_) vs. temperature (T). For the corresponding CD spectra of IF2 shown in (**a**) at 4, 40, and 95 °C, see Appendix A, Appendix A. Data were normalized based on the averaged signal measured when the protein is folded (between 35 and 45 °C) and to the U state based on the average signal measured when the protein is unfolded (>90 °C, for fitting see Appendix A, Appendix A). (**c**) The thermal denaturation of IF2 in 0.1–2 M NH_4_Cl, NaCl, CsCl, LiCl, and NaClO_4_ (*T_trs, cold_* are labeled in gray and *T_trs,_
_hot_* in black).

**Figure 3 ijms-24-06787-f003:**
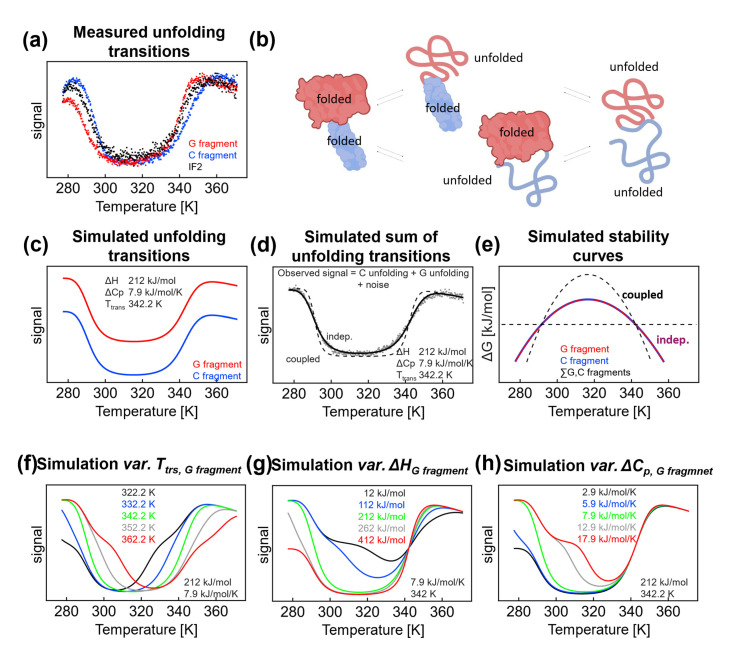
(**a**) The thermal denaturation of the G fragment (red), C fragment (blue), and full-length IF2 (black) measured by circular dichroism at 222 nm. (**b**) Two denaturation pathways of IF2: in one pathway, denaturation starts with the C fragment, and in the second pathway, denaturation starts with the G fragment. When the fragments have equal stability and are independent, then both pathways are present. (**c**) Independent unfolding of G and C fragments in which transitions are described by the same thermodynamic parameters. (**d**) Simulation of unfolding transition of IF2 in the case when the denaturation of both G and C fragments is coupled (dashed line) and independent (solid line) and measured data (gray dots). In simulations, data are described by the same parameters. (**e**) Simulated stability curves for independent G/C fragments (solid red and blue line) and coupled fragments (dashed black line) described by the same thermodynamic parameters as in (**d**). Simulation of changes in the thermal unfolding profile when the denaturation parameters for the C fragment are held constant (Δ*H* 212 kJ/mol/K, Δ*c*_p_ 7.9 kJ/mol, *T_t_*_rs*, hot*_ 342.2 K) while one of the parameters for the G fragment is varied (**f**) variable *T*_trs*, hot*_, Δ*H* 212 kJ/mol/K, and Δ*c*_p_ 7.9 kJ/mol and (**g**) *T*_trs*, hot*_ 342.2 K, variable Δ*H*, and Δ*c*_p_ 7.9 kJ/mol. (**h**) Simulated effect of changes in the values of the heat capacity for the G fragment with the other values held constant. See also Appendix A, Appendix A.

**Figure 4 ijms-24-06787-f004:**
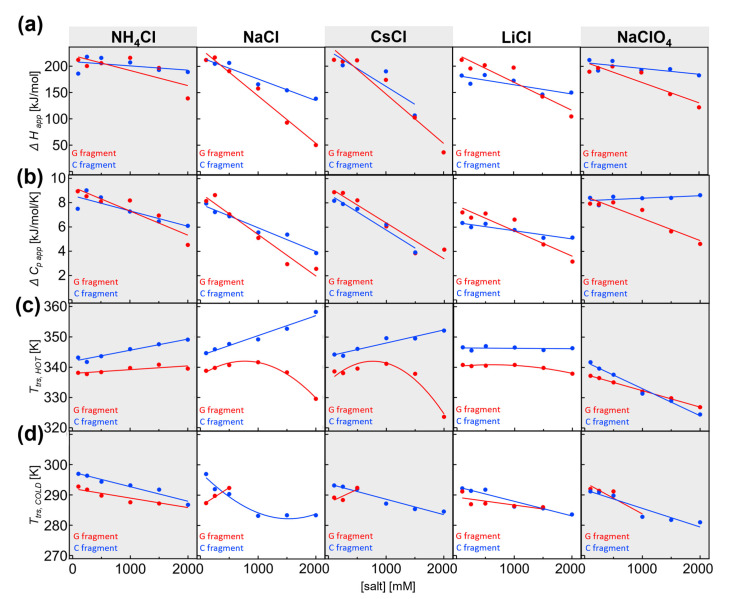
The thermodynamic parameters obtained from nonlinear regression of the cold and hot denaturation data: Δ*H* (**a**); Δ*c_p_* (**b**); *T_trs, hot_* (**c**); and (**d**) *T_trs, cold_* for G fragment (red) and C fragment (blue) in the presence of 9 M urea, and 0.1–2 M salts (NH_4_Cl; NaCl; CsCl; LiCl; and NaClO_4_). Data were analyzed using linear regression analysis, and Pearson correlation coefficients are listed in Appendix A. In addition, the table contains the percentage of the probability *Prob*_N_ (|*r*| ≥ *r*_0_) that the correlation coefficient was observed for uncorrelated variables containing *N* data pairs (see [33]). The quadratic fit was applied when the residuals from linear regression analysis displayed significant trends, deviations, and/or serial correlations. The major function of the regression fits was to guide the eye and quantify the salt sensitivity of the given parameter. For residual plots, see [34,35].

## Data Availability

Data are available on request by authors.

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
