# Peer review of "Salt-Specific Suppression of the Cold Denaturation of Thermophilic Multidomain Initiation Factor 2"

_ijms, 2023, doi:10.3390/ijms24076787_

Round 1
Reviewer 1 Report
The authors present an experimental study of hot and cold denaturation of a translation initiation factor from a thermophilic organism.
I found the results somehow interesting, as they allow to discuss the subtleties of cold denaturation, most often overlooked. However, the clarity of the presentation by the authors needs to be improved, while research design and data analysis also need to be improved (see in particular point 10). Finally, several statement suffers from a poor english writing (a non exhaustive list is: lines 36-39, line 77, lines 372-373, lines 434-438, line 473, lines 524-525)
Clarity presentation issues
1) introduction: it is not clear how many cryo-em models of IF2 the authors are referring to; they first mention one from E Coli, than they show one in Fig. 1 from Thermus Thermophilus (although I am not completely sure about that either). This needs to be stated clearly in the main text as well.
2) number of domains in the G fragment: Fig. 1 shows 2 domains (G1 an G2); however in the final 3.3 section the authors refer to G2/G3 domains and explicitly state that the G fragment has a larger number (that is > 2) than the C fragment. This point needs to be clarified.
3) lines 328-329: I guess G and C in this sentence need to be exchanged
4) lines 342-343 and 360-361: the authors state twice that they performed a simulation of the individual denaturation transitions for the G and the C fragments. What do they mean? What results do they get? Where are such results shown/discussed?
5) The discussion at lines 374-378 is confusing. First, it is not clear when they discuss the behavior of the enthalpy and when the one of the specific heat change (the use of "such strong dependence" is particularly confusing). Second, referring to the specific heat change, I would say from figure 3 that the only clear selectivity is for anions for the C fragment whereas for cations (comparing NaCl and NaH4Cl at least) results look similar for both fragments. Unless the authors refer to the LiCl case (but they should state this clearly).
6) The discussion of why the authors perform their study using urea is confusing. At the end of the introduction section, the authors state that urea is needed to prevent aggregation at high T of the C fragment, and thus to allow the study of a reversible thermal denaturation. In the discussion section (3.1) they instead state that urea is needed as a condition for cold denaturation. This prompts a couple of questions that the authors ought to answer: do they have data for low T in the absence of denaturants? If yes why are such data not shown? Also, the overall motivation of studying cold denaturation for thermophilic proteins in view of industrial applications (which could be spoiled by cold denaturation occurring at room temperature) is somehow weakened: you just do not add urea and you do not have this issue anymore. Of course I agree with the academic interest of studying cold denaturation in thermophiles.
7) discussion section 3.2; at lines 500-506 the authors use the term "denaturation mechanism" in what I think could be a misleading way. I realize they refer to the presence or lack of cooperative denaturation, for example for different domains in a multi-domain protein (and I appreciate their conclusion on the need of caution in interpreting experimental data due to limited resolution). Yet, at first reading I was rather thinking to different mechanisms driving cold and heat denaturation in terms of interactions between water molecules and hydrophobic groups; in this respect, the two denaturations obviously have different mechanisms. I suggest the authors to clarify what they mean by mechanism.
DATA ANALYSIS & RESEARCH DESIGN
8) The authors need to explain in detail how the fits in Figure 4 are performed as a function of salt concentration. In particular, upon which conditions do they choose to perform a linear or (seemingly) quadratic fit? For example, by comparing LiCl data in b) and c) the fact that a quadratic fit is used in c) and a linear fit is used in b) seems arbitrary to me.
9) in the discussion section 3.3, at lines 539-540, the authors state that cold denaturation is not shifted towards higher temperatures. Since they are discussing the salt data in Figure 4 (for perchlorate) I do not really understand what they are referring to. T_trs (eq. 2) is the heat denaturation temperature and in fig. 4 no reference is made to any cold denaturation temperature. Actually, I would strongly suggest the authors to consider computing T_cold given eq. 2 and to show and discuss the results for all the salt concentrations.
10) In the result section 2.3, lines 358-360, the authors comment that the difference in melting temperatures between the two fragments for high concentration of CsCl is high enough (almost 30 C) to be potentially observable in the full length IF2. In fact, they already performed the experiment (Fig. 2C I guess) and, apparently, they do not observe any double melting transition. This seems to me a crucial weakness in the authors approach. Similarly, the authors should compute T_cold dependence on concentration for all salts (see point 9 above) and find if in some case the two fragments display a high enough difference and, if true, compare this prediction with the experiment for the full length protein.
Reviewer 2 Report
The authors have investigated Salt-specific suppression of the cold denaturation of thermophilic multi-domain initiation factor 2. This is a well-written manuscript and the messages are clear. I will go with the publication in its current form.
Reviewer 3 Report
In the present manuscript, the authors investigate the hot and cold denaturations of a thermophilic protein in the presence of 9 M urea. The exploit different salts to investigate the chaotropic Vs kosmotropic effect. Then they investigate two separate domains to get insight into the unfolding mechanisms. The authors find that salts display specific effects on cold denaturation, while most of the salts increase melting temperature (except for perchlorate).
The manuscript may be interesting and relevant to scientists working on the denaturation and the stability of thermophilic proteins. However, I found some problems that should be considered prior to publications.
MAJOR COMMENTS
. My main concern involves the possibility to dissect kosmotropic and chaotropic effects with an experimental setting that exploits 9 M urea, which is chotropic itself, and the authors use it exactly for that purpose. 9 M is a very high concentration, and it is very difficult to imagine that water would behave the same way as it would in the absence of such strong denaturant. Would the results change exploiting GndHCl or GndSCN as denaturants? I think the authors should comment in the text on these problems.
. The methods section of the manuscript is honestly unacceptable, among the shortest I have ever read. I understand that brevity is important, but this is too much. How were the proteins prepared and purified? How were the fits carried out? Did Sf and Su in eq. (1) reflect a constant value or a linear dependence upon increasing temperatures? For example, fig. 3 c seems to show a theoretical trend in which the signal of U decreases at high temperature. Where does eq. (1) come from?
. Authors normalized experimental data in Figure 2b so that the figure shows the unfolded fraction. However, the normalization seems to be flawed as it was probably carried out using the lowest and highest experimentale values. In this way the unfolded frction seems never to be 0. It would be more advisable to normalize data assigning to the F state the average signal measured when the protein is folded (e.g. between 35 and 45 degrees), and to the U state the average signal measured when the protein is unfolded (e.g. above 90 degrees). This observation is not irrelevant with respect to the manuscript development, as the authors use this figure to state that the protein is not completely folded at room T.
MINOR COMMENTS
. Figure 3 has a very poor definition. This should be improved in the final version of the manuscript.
. Figure 3 h reports temperature in °C but the x axis label reports [K]
. line 577: "at a constant scan rate of 1 °C/min". Perhaps the authors meant "at a constant temperature gradient of 1 °C/min"?
. Eq. (1) is formally incorrect. The notation exp usually requires the argument "in-line"; instead, when one uses "e" then the exponent goes to te exponent.
. 588: The enthalpy change in eq. (2) is the enthalpy change at Ttrs, using authors' notation. Authors should change the symbols to make this clear.
Reviewer 4 Report
Džupponová and co-authors analyzed the denaturation of IF2 in thermophilic bacteria using CD spectra. Sufficient CD spectral data are provided to show temperature-dependent refolding and unfolding of IF2. These data are analyzed thermodynamically, and the structural stabilization by salts is eagerly discussed. On the other hand, the authors do not provide any experimental evidence other than CD spectra, and I am somewhat concerned about whether they are convincing enough to be published in IJMS. The credibility of this paper may be enhanced if the authors provide sufficient explanations or interpretations based on previous studies on the points I raise below.
Line 148: If the experiment had started at 4 °C, the sample would have been unfolded according to the authors. Normally, when large proteins are denatured, irreversible aggregation occurs, making it difficult to control the experimental conditions. In the case of the authors' experiments, the presence of 9 M urea avoided this problem, but if one proceeds with this particular experimental system (even if it has been described in previous papers), the advantages, disadvantages, and limitations should first be briefly explained.
I am also concerned that urea and other salts may precipitate at 4°C and that high concentrations of salts may interfere with the observation of CD spectra. I believe that by presenting the High Tension (HT) Voltage at each temperature simultaneously, the authors can show that their experiments are properly performed.
In addition to these, I would like to see the full spectra presented, for example, under the conditions of Figure 2a, at low temperature denaturation, at steady state, and at high temperature denaturation. Then it will be less misleading and more clear to me and other readers what range of "denaturation" the authors are discussing in the overall IF2 structure.
Line 192: It is not clear to me what purpose Cs and Li are being used for. Mentioning the position of these cations in relation to NH4+ and Na would make the authors' point clearer.
Line 292: So what should we get from SI Materials, which doesn't even have a LEGEND on it?
Figure 4a: For the G fragment, I do not see that ΔH decreases linearly in the presence of NH4Cl, CsCl, and NaClO4; the same is true in the presence of CsCl for the C fragment. If the authors are convinced that, theoretically, there should be a linear decrease, they should check the reproducibility and reduce the error on linearity.
Line 413: I am a little confused by the statement "On the one hand, sodium chloride was very effective in suppressing cold denaturation, and we found that other salts, both kosmotropic and chaotropic, were much less effective." In Figure 2c, a similar structural stabilization was observed for CsCl as for NaCl, but I wonder if the authors are sure that the Cs ions contribute nothing at all.
Line 417: I would like to know what properties of the salts the authors believe are responsible for the different stabilization or destabilization that different salts exhibit. Are electronegativity, ionic radius, etc. sufficient to explain the salt effect?
Line 503: I do not follow the experiments in Ref [39], but at least what is observed in this paper is "refolding" at low temperatures and unfolding at high temperatures. I do not believe that the refolding and unfolding processes are completely equivalent under either low-temperature or high-temperature conditions. I would like, if possible, for the authors to provide a basis for assuming that the refolding and unfolding processes are equivalent (at least under the conditions of their experiment). In Ref [28], did the authors confirm that the CD profile does not change after repeated temperature transition experiments on the same sample?
Figure 5e: This figure is not friendly to me at all. I have not received any information about the purpose of frustration analysis and what the green and red lines mean.
Line 565: Does this ALL chemicals include proteins? Write down the catalog numbers of salts and proteins that are important in this paper. How did the authors prepare the IF2 fragments? I am willing to trust the authors' data by obtaining these pieces of information clearly.
Line 580: At what temperature did the authors start the temperature transition experiment? I recomend the Method Section should also clearly state whether they started from an unfolded state of 4°C or whether they scanned from a folded state at room temperature to the high and low temperature side.
Minor points
Figure 2b: I cannot distinguish between the data points because they overlap in some areas. I suggest that each type of salt should be indicated with a different color, marker, or saturation.
Figure 2c: Units should be clearly indicated.
Line 221: I can believe the claim that "the authors have previously shown", but the appropriate reference should be clearly indicated.
Figure 5d: I recognize that blue is a positively charged surface and red is a negatively charged surface, but the authors should clarify in the figure what the colors indicate.
Round 2
Reviewer 1 Report
The authors made a great effort in revising the manuscript. As a result it is much improved, as they successfully addressed all of my comments. It was not simple to read the "track-changes-on" revision, given their number. There are a couple of sentences to be revised
- lines 190-191
"Both denaturing transitions are cooperative, and the temperatures corresponding to cold denaturation are T_cold = 18 \pm 0.4 °C and T_hot = 67.4 \pm 0.2 °C." --> "... corresponding to cold AND HOT denaturation are ....."
- line 467
"As a part of the full-length IF2, the T_hot of fragments differ by less than 10.5 °C,; hence, we could resolve the individual transitions."
--> "... we could NOT resolve ..." (I guess)
Author Response
"Please see the attachment."

Reviewer 3 Report
The authors replied to all my comments. I have no further objections to publication of the manuscript.
Author Response
"Please see the attachment."

Reviewer 4 Report
The authors have responded very sincerely to the points I have raised, and the presentation of the results has been greatly improved. In my opinion, the current form is acceptable for publication in terms of the quality of the data and its description.
If possible, in the proof, the chmicals should be presented with a catalog number instead of a CAS No. The catalog number gives us important information on the purity and properties of the reagents, and improves the reproducibility.
Ex) sodium cacodylate trihydrate 776 (CAS No. 6131-99-3)
-> (Cat. No. C0250) or (Cat. No. C4945)
Author Response
"Please see the attachment."
